

# Identification of keygenes, miRNAs and miRNA-mRNA regulatory pathways for chemotherapy resistance in ovarian cancer

Wenwen Wang[1,2], Wenwen Zhang[3,4] and Yuanjing Hu[3,4]

[1] Tianjin Medical University, Tianjin, China
[2] Department of Obstetrics and Gynecology, Beijing Tongren Hospital affiliated Capital Medical University, Beijing, China
[3] Department of Gynecological Oncology, Tianjin Central Hospital of Obstetrics and Gynecology, Tianjin, China
[4] Department of Gynecological Oncology, Obstetrics and Gynecology Hospital affiliated Nankai University, Tianjin, China

## ABSTRACT

**Background:** Chemotherapy resistance, especially platinum resistance, is the main cause of poor prognosis of ovarian cancer. It is of great urgency to find molecular markers and mechanism related to platinum resistance in ovarian cancer.

**Methods:** One mRNA dataset (GSE28739) and one miRNA dataset (GSE25202) were acquired from Gene Expression Omnibus (GEO) database. The GEO2R tool was used to screen out differentially expressed genes (DEGs) and differentially expressed miRNAs (DE-miRNAs) between platinum-resistant and platinum-sensitive ovarian cancer patients. Gene Ontology (GO) function and Kyoto Encyclopedia of Genes and Genomes (KEGG) pathway enrichment analysis for DEGs were performed using the DAVID to present the most visibly enriched pathways. Protein–protein interaction (PPI) of these DEGs was constructed based on the information of the STRING database. Hub genes related to platinum resistance were visualized by Cytoscape software. Then, we chose seven interested hub genes to further validate using qRT-PCR in A2780 ovarian cancer cell lines. And, at last, the TF-miRNA-target genes regulatory network was predicted and constructed using miRNet software.

**Results:** A total of 63 upregulated DEGs, 124 downregulated DEGs, four upregulated miRNAs and six downregulated miRNAs were identified. From the PPI network, the top 10 hub genes were identified, which were associated with platinum resistance. Our further qRT-PCR showed that seven hub genes (BUB1, KIF2C, NUP43, NDC80, NUF2, CCNB2 and CENPN) were differentially expressed in platinum-resistant ovarian cancer cells. Furthermore, the upstream transcription factors (TF) for upregulated DE-miRNAs were SMAD4, NFKB1, SMAD3, TP53 and HNF4A. Three overlapping downstream target genes (KIF2C, STAT3 and BUB1) were identified by miRNet, which was regulated by hsa-miR-494.

**Conclusions:** The TF-miRNA–mRNA regulatory pairs, that is TF (SMAD4, NFKB1 and SMAD3)-miR-494-target genes (KIF2C, STAT3 and BUB1), were established. In conclusion, the present study is of great significance to find the key genes of platinum resistance in ovarian cancer. Further study is needed to identify the mechanism of these genes in ovarian cancer.

Corresponding author
Yuanjing Hu, hyj_tdj@sina.com

# INTRODUCTION

Epithelial ovarian cancer (EOC) is the third highest incidence of carcinoma in the female reproductive system with the highest mortality rate. The standard of treatment of ovarian cancer is surgical resection followed by platinum-based chemotherapy (*Armstrong et al., 2021*).

Platinum compounds bind to DNA and create intra-strand or inter-strand DNA crosslinks that induce DNA damage (*Wang & Lippard, 2005*). However, chemotherapy resistance, especially platinum resistance, leads to poor prognosis of ovarian cancer patients (*Agarwal & Kaye, 2003*). However, the molecular mechanisms of platinum resistance in ovarian cancer remain unclear. Therefore, a comprehensive understanding of the mechanism of platinum resistance is the key to improve outcomes of women with EOC.

Currently, despite more and more studies use bioinformatics methods to analyze differentially expressed genes and miRNAs in ovarian cancer (*Yang et al., 2020*; *Zheng et al., 2019*; *Cao et al., 2019*; *Feng et al., 2019*; *Gong, Lin & Yuan, 2020*; *Li & Li, 2019*; *Lu et al., 2020*; *Zhang et al., 2019*), to our knowledge, a systematic and comprehensive analysis of miRNA-mRNA regulatory network based on clinical samples of cisplatin resistance in ovarian cancer is still absent. In this study, we tried to construct and analyze the transcription factors-miRNA–target genes regulatory network using bioinformatics methods in order to find the potential mechanism and markers of platinum resistance in ovarian cancer patients. To detect the DEGs and DE-miRNAs between platinum-resistant and platinum-sensitive ovarian cancer patients, we used the datasets downloaded from the Gene Expression Omnibus (GEO) database. Gene Ontology (GO) functional annotation analysis and Kyoto Encyclopedia of Genes and Genomes (KEGG) pathway enrichment analysis were performed using the DAVID for the screened DEGs. We established a protein–protein interaction (PPI) network to identify hub genes related to platinum resistance. Then, the analyzed results were verified by using quantitative reverse transcription polymerase chain reaction (qRT-PCR) assay in A2780 ovarian cancer cells. At last, the TF-miRNA-target genes regulatory network was predicted and constructed using miRNet software and Cytoscape.

# MATERIALS & METHODS

## Data source

The gene expression datasets analyzed in this study were obtained from the GEO database (https://www.ncbi.nlm.nih.gov/geo/). GSE28739 was established on the platform of GPL7264 (Agilent-012097 Human 1A Microarray (V2)). There were 25 patients in this data set, including 15 chemo-resistant serous EOC patients (time to recurrence (TTR) ≤ 6 months) and 10 chemo-sensitive serous EOC patients (TTR ≥ 30 months). A total of

30 chemo-resistant serous EOC samples and 20 chemo-sensitive serous EOC samples were analyzed to screen DEGs.

GSE25202 was based on the platform of GPL8179 (Illumina Human v2 MicroRNA expression beadchip; Illumina, San Diego, CA, USA). The dataset contained 55 surgical specimens including 30 early relapsing patients (optimally debulked patients with TTR < 12 months and sub-optimally debulked patients with TTR < 6 months) and 25 late relapsing patients (optimally debulked patients with TTR > 36 months and sub-optimally debulked patients with TTR > 12 months). All of the data were available for free online.

### Identification of DEGs and DE-miRNAs

GEO online analysis tool GEO2R (https://www.ncbi.nlm.nih.gov/geo/geo2r/) was used. Genes that met the criteria (adjusted $p$ value < 0.05 and |$\log_2$Fold Change| ≥ 0.5) were considered as DEGs. Adjusted $p$ value < 0.05 and |$\log_2$Fold Change| ≥ 1.5 were set as the thresholds for identifying DE-miRNAs.

### GO functional enrichment analysis of DEGs

GO functional analysis is a common method to analyze the functions of DEGs,which can be classified into biological process (BP), molecular function (MF), and cellular component (CC). GO functional analysis was carried out utilizing DAVID online tool (https://david.ncifcrf.gov/). $p$ value < 0.05 and gene counts ≥ 3 were considered as significant.

### KEGG pathway analysis of DEGs

KEGG pathway analysis is a widely used database to explore potential relationships among these DEGs. It was also carried out by DAVID online tool. $p$ value < 0.05 and gene counts ≥ 5 were considered as significant.

### PPI network construction and hub gene identification

The online Search Tool for the Retrieval of Interacting Genes (STRING) database is used to construct PPI network among DEGs (http://string-db.org/). Subsequently, the PPI network was visualized by Cytoscape software (www.cytoscape.org/). One plugin in Cytoscape, CytoHubba, was used to screen hub genes, which were evaluated by connectivity degree of each protein node. The top 10 genes were identified as hub genes in the study.

### Prediction of potential transcription factors and target genes of DE-miRNAs

The upstream transcription factors (TF) and the target genes of the DE-miRNAs were predicted using miRNet database (https://www.mirnet.ca/). The TF-miRNA-target genes regulatory network was depicted and visualized using Cytoscape software (www.cytoscape.org/).
**Table 1 Nucleotide sequences of the primers.**

| Gene name | Forward primer (5′ to 3′) | Reverse primer (5′ to 3′) |
| --- | --- | --- |
| NUF2 | GGAAGGCTTCTTACCATTCAGC | GACTTGTCCGTTTTGCTTTTGG |
| CCNB2 | CCGACGGTGTCCAGTGATTT | TGTTGTTTTGGTGGGTTGAACT |
| CENPN | TGAACTGACAACAATCCTGAAGG | CTTGCACGCTTTTCCTCACAC |
| KIF2C | CTGTTTCCCGGTCTCGCTATC | AGAAGCTGTAAGAGTTCTGGGT |
| NUP43 | TGCCTCCGGGAAGTTTACAGA | TCTCCTTCAAACCCTCCATCA |
| NDC80 | CCTCTCCATGCAGGAGTTAAGA | GGTCTCGGGTCCTTGATTTTCT |
| BUB1 | TGGGAAAGATACATACAGTGGGT | AGGGGATGACAGGGTTCCAAT |
| GAPDH | GGAGCGAGATCCCTCCAAAAT | GGCTGTTGTCATACTTCTCATGG |

## Drug preparation

Cis-Diammineplatinum dichloride (Cis-platinum) was obtained from Aladdin (Shanghai, China) and dissolved in Dimethyl sulfoxide (DMSO) at a final concentration of 2 mg/ml. Aliquots were stored at −20 °C and thawed immediately before use.

## Cell lines

A2780 and its cis-platinum resistant cell lines were gifts from Professor Li Bin from National Cancer Institute, Cancer Hospital, Chinese Academy of Medical Sciences, Peking Union Medical College. A2780 cells were maintained in RPMI-1640 containing 10% fetal bovine serum under atmosphere of 5% $CO_2$ with humidity at 37 °C. A2780 cis-platinum resistant cell line was cultured in RPMI-1640 containing cis-platinum with the concentration of 200 ng/ml and 10% fetal bovine serum under atmosphere of 5% $CO_2$ with humidity at 37 °C.

## Quantitative RT-PCR

Total RNAs from cell lines were extracted using TRIzol Reagent (Ambion, Austin, TX, USA) according to the manufacturer's instructions. Total RNA was then reverse transcribed into cDNA in a final volume of 40 μL using M5 Super qPCR RT kit (Mei5 Biotechonlogy, Beijing, China). Then qPCR amplifications were carried out in the 20 μL reaction mixtures with 2 × SYBR Green qPCR Master Mix (Bimake, Houston, TX, USA). The program was as follows: 50 °C for 2 mins, 95 °C for 10 mins, 95 °C for 15 s and 60 °C for 1 min for 45 cycles. The primers were purchased from Sangon Biotech Co. (Shanghai, China). Expression of GAPDH was used as the internal control, and the nucleotide sequences for primers are listed in Table 1. The relative mRNA levels were analyzed by the $2^{\Delta\Delta CT}$ method. Each experiment was performed in triplicate.

## Statistical analysis

Statistical analysis was performed using SPSS 22.0. Student's t test was used to assess the significance among groups. GraphPad Prism 8.0.2 was used for plotting graphs. $p < 0.05$ was considered to be a significant difference.

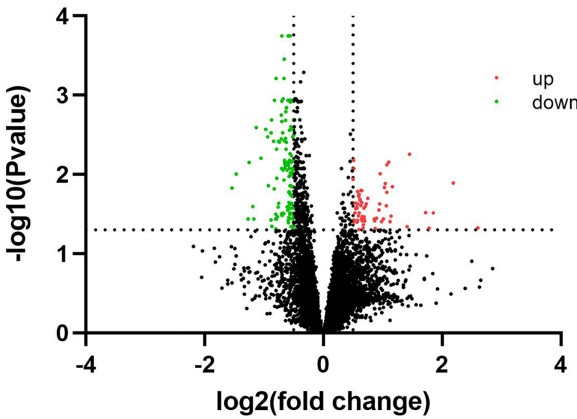

**GSE28739 resistant vs sensitive**

**Figure 1 Volcano plot of differentially expressed genes (DEGs).** Red: upregulated genes; Green: downregulated genes.               

**Table 2 Information about the DEGs identified from the datasets.** A total of 187 DEGs were identified from GSE28739, including 63 upregulated DEGs and 124 downregulated DEGs in platinum-resistant ovarian cancer samples compared to platinum-sensitive samples.

| DEGs | Gene name |
|---|---|
| Upregulated (63) | THEMIS2, ACADVL, EGR2, HBA2, HBB, HBA1, HBG1, IL1B, COMT, PTGDS, EVI2A, SREBF1, INPP5D, EHBP1L1, ARHGDIB, IL10RA, GRB2, IGFBP6, PLEK, CCL2, HIRA, SOCS3, FCGR2B, PLK3, CD163, ARHGDIA, DUSP2, DUSP5, OVOL1, CEBPB, CD164, NNMT, SMPDL3A, LAPTM5, PNRC1, IFI44L, VSIG4, CDC42EP4, DPP7, MICAL2, SWAP70, STAB1, ABI3BP, CPAMD8, RMDN1, FCGR2A, DIDO1, IFI6, METRN, NFKBIZ, GNPTG, HAVCR2, CARD9, CMTM7, SCEL, OSR1, ZNF385B, COG7, CCDC117, C1orf162, ARID5A, STAT3, MALAT1 |
| Downregulated (124) | FBXW4P1, RBM33, SLC25A16, MSH6, MID1, GSTP1, SSBP3, LAMB3, MYO19, C2orf88, ZBTB38, ASRGL1, GBAS, ASNS, CKS2, COX7B, HSPE1, EPCAM, NDUFB3, POLA2, POU2F1, SCG5, SNRPE, SRP19, SSBP1, ZNF45, HIST1H4I, HIST1H2BN, HIST1H2BE, HIST1H2BC, HIST1H4F, HIST1H4C, HIST1H4H, HIST1H4B, HIST1H4L, LGR5, STC2, MPZL1, FKBP5, BUB1, COX6A1, GCSH, NDUFA1, PSPH, CCNB2, C14orf2, SFPQ, HINT1, LAMC2, SMO, TCEB1, RABEPK, POP7, PRSS16, ACAT2, NDC80, SIVA1, EBNA1BP2, KIF2C, PYCR1, MYCBP, PRDM4, H2AFV, DNAJC15, HILPDA, METTL5, CRIPT, CNIH4, SEC61G, FAM162A, MELK, ZNF777, TRIM24, MRPS17, MRPS33, SCFD1, CHCHD2, TRIAP1, RSRC1, WAC, FOLR1, F11R, C1orf109, YEATS2, STYK1, CENPN, LRRC59, ZNF395, PCDHB2, EML4, SPC25, SYT13, ZNF14, RAB38, SOX17, TMEM106C, TRAPPC6A, UBL5, CDCA7, AJUBA, PURB, SIGLEC11, TSEN15, PSAT1, MACROD2, NSMCE3, TRUB1, CIART, NUF2, GSTA1, SGPP2, CDCA2, ZNF48, TMEM139, GSTA5, NASP, ERICH5, TMEM136, PTPMT1, RFC5, GOLT1A, NUP43, FGF7P6, TP53TG1 |

# RESULTS

## Identification of DEGs and DE-miRNAs

Based on GEO2R analysis, a total of 187 DEGs (63 upregulated genes and 124 downregulated genes) were identified based on the criteria of adjusted $p$ value < 0.05 and $|\log_2$Fold Change$| \geq 0.5$. The volcano plot of DEGs is shown in Fig. 1. All of the upregulated and downregulated DEGs are shown in Table 2. In miRNA datasets GES25202, a total of 10 DE-miRNAs (four upregulated miRNAs and six downregulated miRNAs) were detected, which was shown in Table 3. The volcano plot of miRNAs is presented in Fig. 2.

**Table 3 Information about the DE-miRNAs identified from the datasets.** A total of 10 DE-miRNAs were identified from GSE25202, including four upregulated miRNAs and six downregulated miRNAs in platinum-resistant ovarian cancer samples compared to platinum-sensitive samples.

| DE-miRNAs | MiRNA name |
|---|---|
| Upregulated (4) | hsa-miR-656, hsa-miR-655, hsa-miR-379*, hsa-miR-494 |
| Downregulated (6) | hsa-miR-506, hsa-miR-513a-3p, hsa-miR-513c, hsa-miR-510, hsa-miR-509-3-5p, hsa-miR-202*:9.1 |

Note:
* miRNA with low expression level.

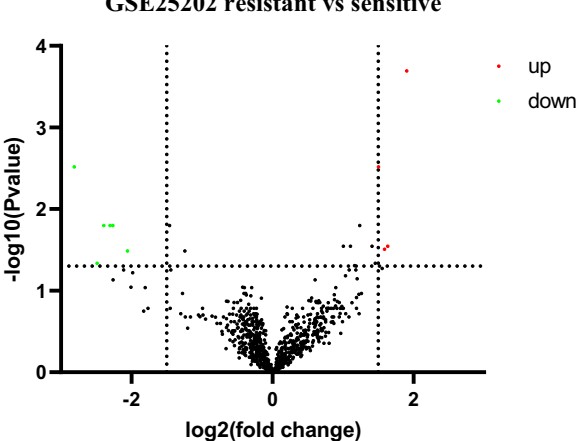

**Figure 2 Volcano plot of differentially expressed miRNAs (DE-miRNAs).** Red: upregulated miRNAs; Green: downregulated miRNAs.

## Gene ontology term enrichment analysis of DEGs

The results of GO functional analysis are shown in Fig. 3. Biological processes (BP) contained nucleosome assembly, CENP-A containing nucleosome assembly, telomere organization, DNA-templated transcription, chromatin, silencing at rDNA, protein heterotetramerization, beta-catenin-TCF complex assembly, negative regulation of gene expression, positive regulation of gene expression, oxygen transport, double-strand break repair *via* nonhomologous end joining and sister chromatid cohesion. For the cellular component category (CC), the DEGs were enriched in nucleosome, nuclear chromosome, hemoglobin complex, condensed chromosome kinetochore, haptoglobin-hemoglobin complex, Ndc80 complex, extracellular exosome. For the molecular function category (MF), the DEGs were correlated with protein binding, haptoglobin binding, oxygen transporter activity.

## Kyoto encyclopedia of genes and genomes pathway analysis of DEGs

The results of KEGG pathway enrichment are shown in Fig. 4. The significant KEGG pathways of DEGs included systemic lupus erythematosus; viral carcinogenesis; alcoholism; malaria; African trypanosomiasis; non-alcoholic fatty liver disease (NAFLD); tuberculosis and mismatch repair.

## Protein–protein interaction network and hub gene identification

The PPI network of the DEGs was constructed with STRING tools. The result of PPI network was displayed in Fig. 5. The PPI network included 184 nodes and 108 edges, and

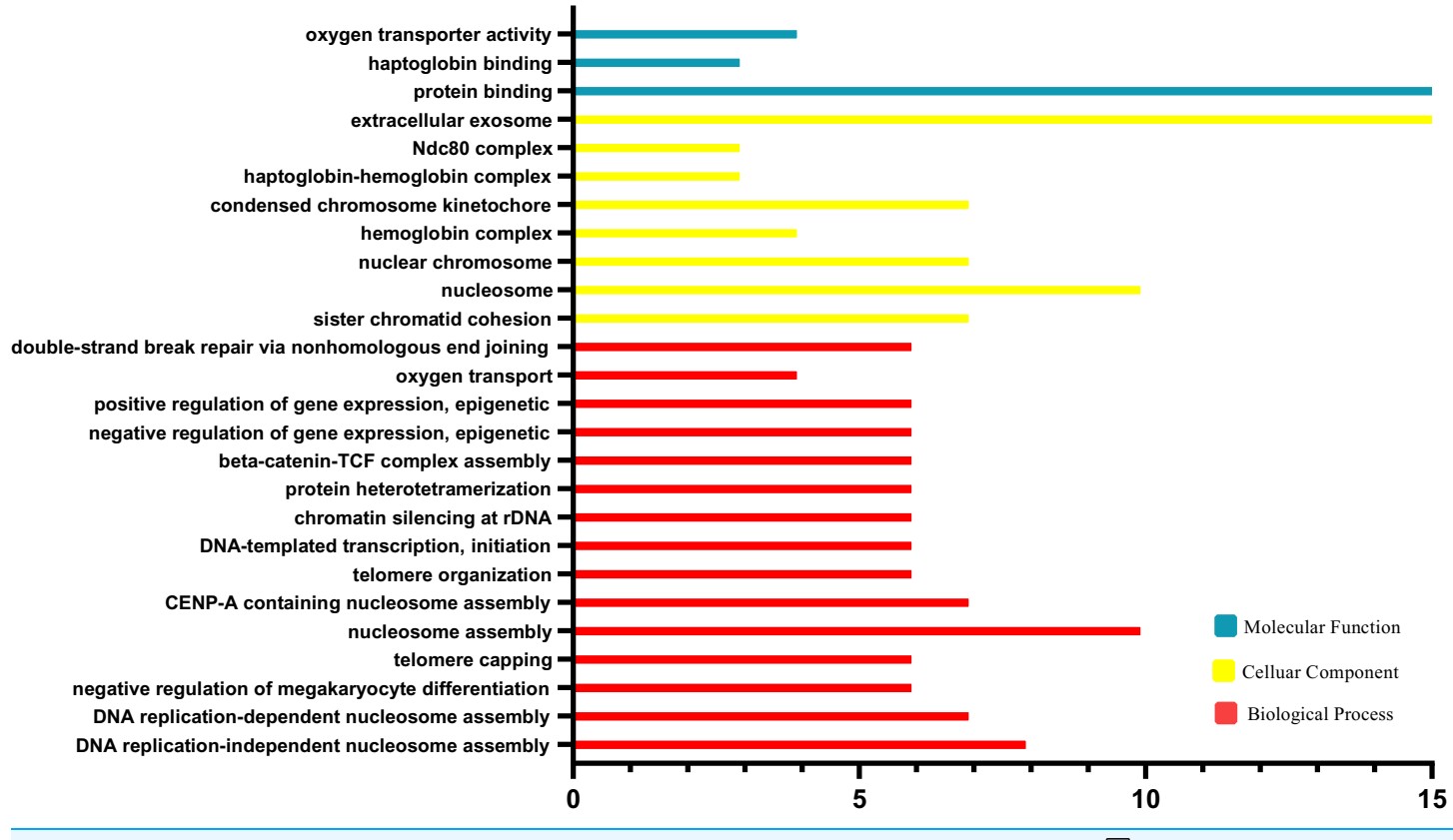

**Figure 3 Geneontology (GO) enrichment analysis.**

the PPI enrichment *p*-value was 1.11e−16. We also structured the PPI network with DEGs and showed hub genes. The top 10 genes (NUF2, CCNB2, CENPN, HIST1H2BE, BUB1, NDC80, NUP43, KIF2C, STAT3 and HIST1H2BC), with the highest degree scores of connectivity in the PPI network were identified as hub genes (Table 4; Figs. 6, 7).

## The expression of hub genes in cell lines

Most of the hub genes are related to the regulation of cell cycle. And we choose seven interested hub genes to validate in A2780 cell lines by qRT-PCR. The results of relative expression of these seven genes are listed in Fig. 8.

## Prediction of transcription factors of DE-miRNAs

In our study, we used miRNet database to predict upstream transcription factors of DE-miRNAs. As all of the 10 hub genes identified in the study were downregulated, so we predicted the transcription factors of upregulated miRNAs, which were presented in Fig. 9. For upregulated DE-miRNAs, the upstream transcription factors were SMAD4, NFKB1, SMAD3, TP53 and HNF4A.

## Construction of miRNA-target gene regulatory network

The miRNet online tool predicted 4,056 target genes for the four upregulated DE-miRNAs. And then, KIF2C, STAT3 and BUB1 among the predicted target genes also belonged to the

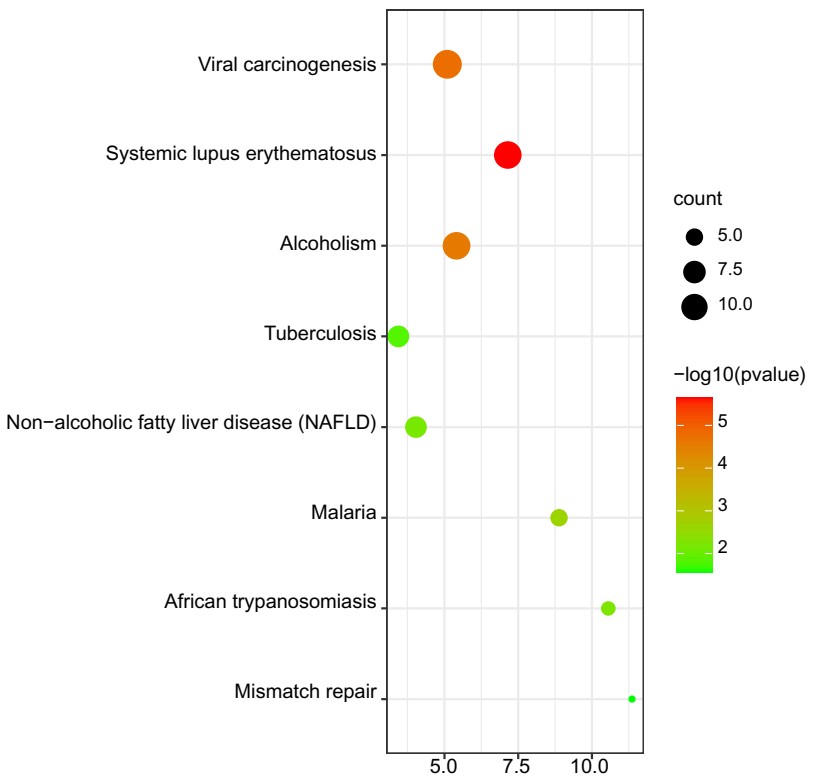

**Figure 4 Bubble map for Kyoto encyclopedia of genes and genomes (KEGG) pathway analysis of DEGs.**

10 hub genes of GSE28739, which was regulated by hsa-miR-494. The miRNA-target gene regulatory network was constructed and visualized in Fig. 10 using Cytoscape.

## DISCUSSION

Clinically, platinum sensitivity is defined as patients with a platinum-free interval (PFI) of greater than 6 months and platinum resistance is defined as those with a PFI of less than 6 months (*Gore et al., 1990*). The reason for platinum resistance in epithelial ovarian cancer is complex. It may include intracellular drug inactivation, reduced intracellular drug accumulation, increased DNA repair and inhibition of cell death (*Freimund et al., 2018*). Currently, mechanisms of platinum resistance in ovarian cancer have been extensively studied, but mostly are in ovarian cancer cell lines (*Dorayappan et al., 2018*; *Mo et al., 2013*; *Tassi et al., 2019*; *Zhang et al., 2018a*; *Huo et al., 2016*; *Zhang et al., 2020*). Hence, considering the heterogeneity of patients, it is crucial to explore the mechanism of platinum resistance in ovarian cancer patients.

MiRNAs are a kind of noncoding RNAs, which bind with targeted mRNAs to regulate mRNA expression. They play a pivotal role in RNA silencing and post-transcriptional regulation of gene expression (*Lee & Dutta, 2009*). However, there have been few studies on miRNAs involved in platinum resistance for ovarian cancer. It may help us identify new markers, specific pathways and targeted therapies for platinum-resistant EOC.

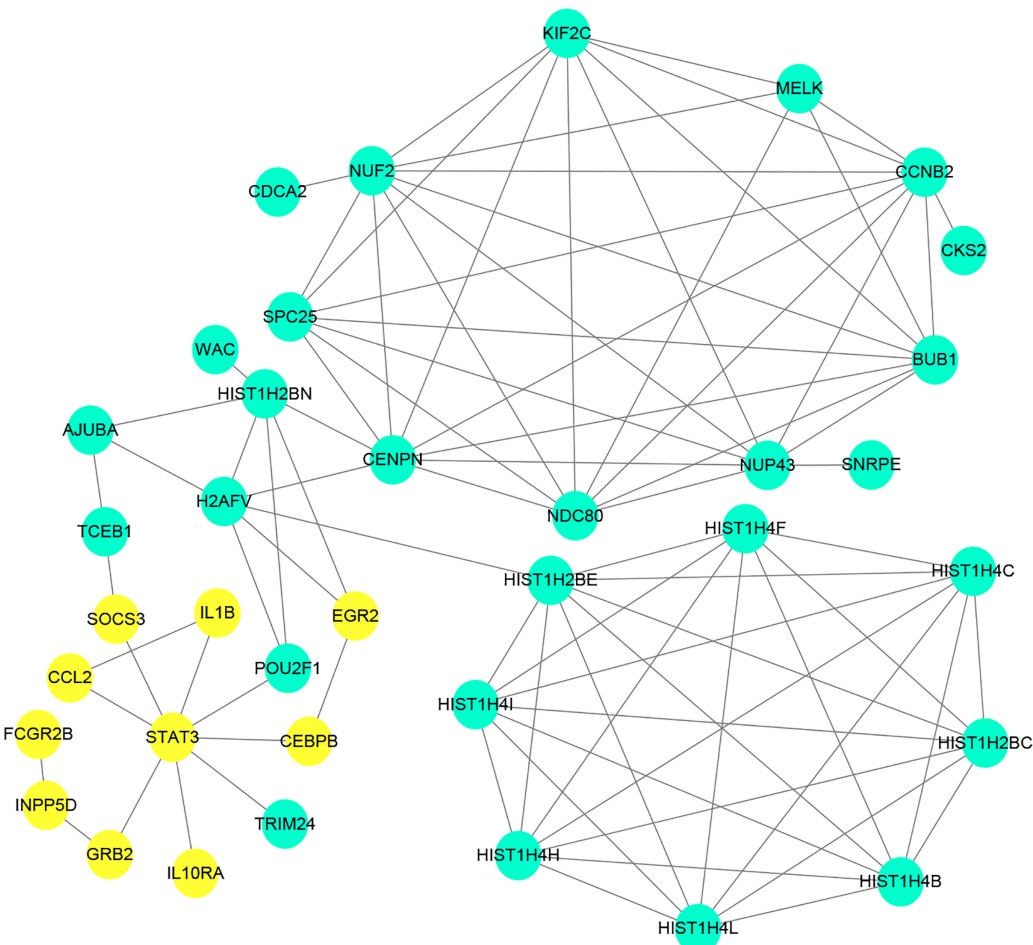

**Figure 5 Protein–proteininteraction network of differentially expressed genes (DEGs).** Circles represent genes, lines represent the interaction of proteins between genes. Yellow: upregulated genes; Blue: downregulated genes.                               

**Table 4 The information about the top 10 hub genes.**

| Gene symbol | Gene description | Score |
| --- | --- | --- |
| NUF2 | NDC80 kinetochore complex component | 9 |
| CCNB2 | cyclin B2 | 9 |
| CENPN | centromere protein N | 9 |
| BUB1 | BUB1 mitotic checkpoint serine/threonine kinase | 8 |
| NDC80 | NDC80, kinetochore complex component | 8 |
| NUP43 | nucleoporin 43 | 8 |
| KIF2C | kinesin family member 2C | 8 |
| STAT3 | signal transducer and activator of transcription 3 | 8 |
| HIST1H2BE | histone cluster 1 H2B family member e | 8 |
| HIST1H2BC | histone cluster 1 H2B family member c | 7 |

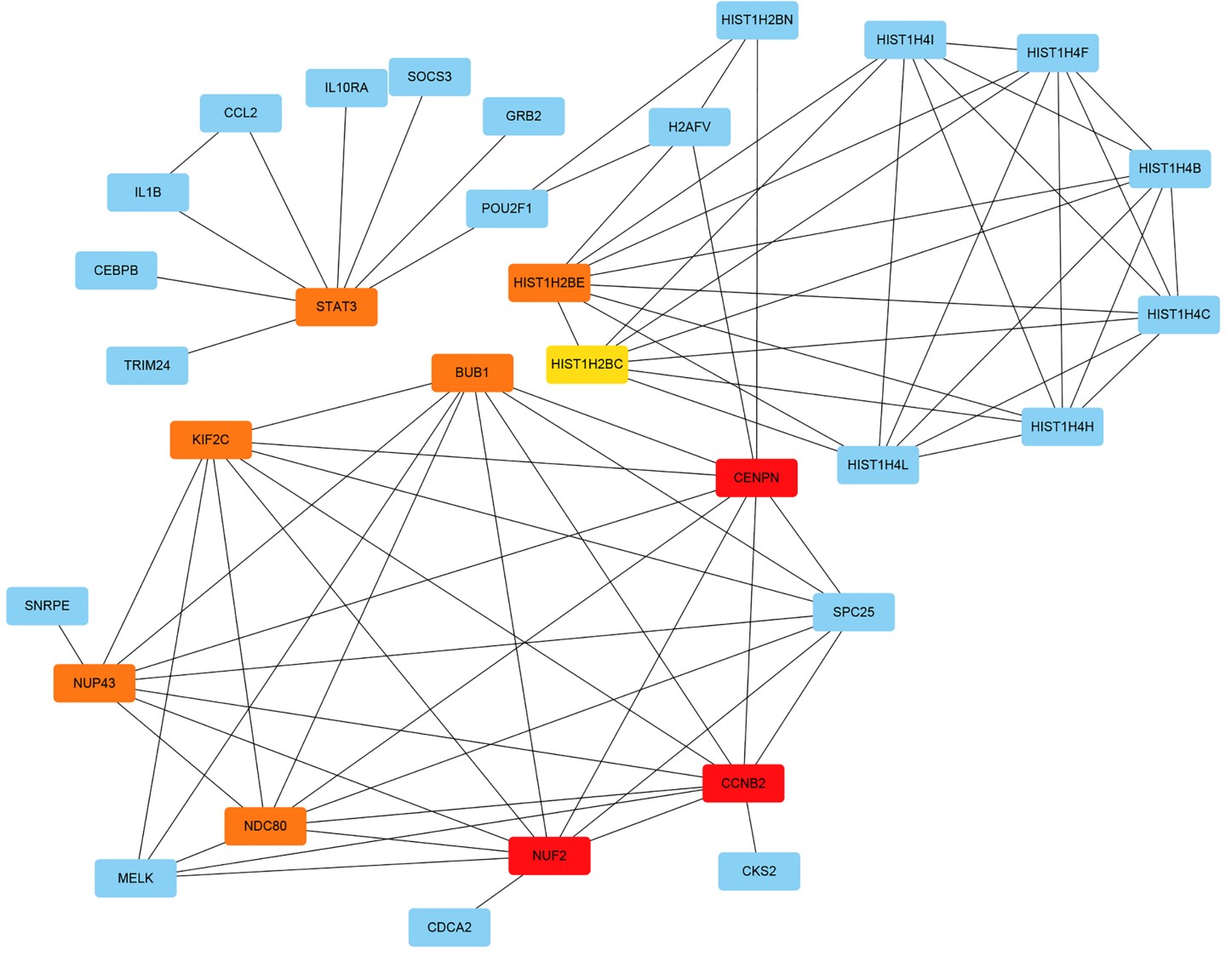

**Figure 6 The interaction network of the hub genes and their related genes.**

In the article, DEGs between platinum-resistant and platinum-sensitive ovarian cancer patients were screened out based on gene expression profiling data from the GEO database. Totally, 63 upregulated DEGs and 124 downregulated DEGs were identified. GO analysis showed that these DEGs were associated with the BP terms, such as nucleosome assembly, CENP-A containing nucleosome assembly, telomere organization, DNA-templated transcription, chromatin, silencing at rDNA, protein heterotetramerization, beta-catenin-TCF complex assembly, negative regulation of gene expression, positive regulation of gene expression, oxygen transport, double-strand break repair *via* nonhomologous end joining and sister chromatid cohesion. The pathways significantly enriched in KEGG analysis contained systemic lupus erythematosus; viral carcinogenesis; alcoholism; malaria; African trypanosomiasis; non-alcoholic fatty liver disease (NAFLD); tuberculosis and mismatch repair. A PPI network was constructed to

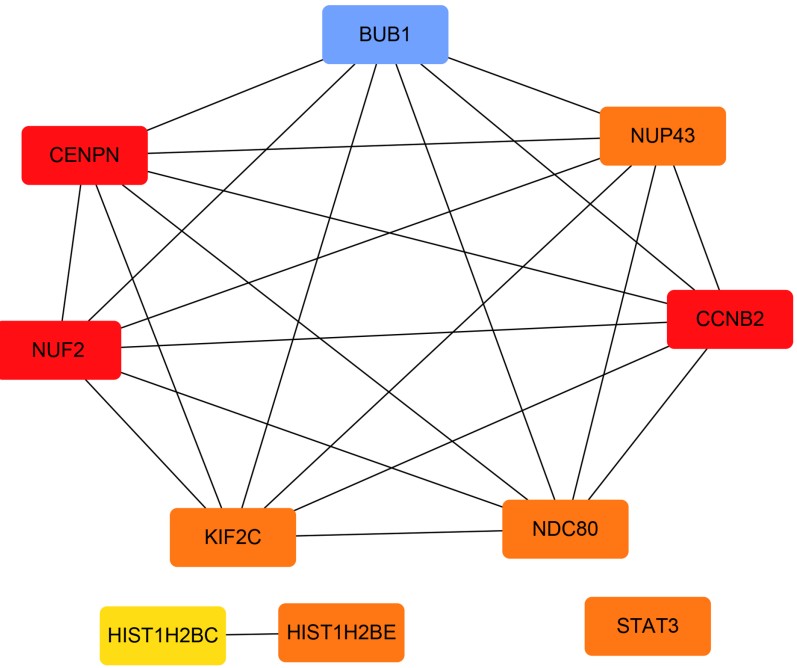

**Figure 7** The hub genes with the highest degree scores of connectivity in the PPI network.

**Figure 8** QRT-PCR analysis of BUB1, KIF2C, NUP43, NDC80, NUF2, CCNB2 and CENPN in A2780 cis-platinum resistant celllines compared with A2780 cell line.

investigate the interrelationship of the DEGs, and 10 hub genes were identified, including NUF2, CCNB2, CENPN, NDC80, HIST1H2BE, HIST1H2BC, BUB1, NUP43, KIF2C and STAT3. All of these genes were downregulated in platinum resistant ovarian cancer patients.

As all of the 10 hub genes identified in our study were downregulated, four upregulated miRNAs (miR-656, miR-655, miR-379*, miR-494) were finally selected. After conducting a combined analysis of DEGs and target genes of DE-miRNAs, three overlapping

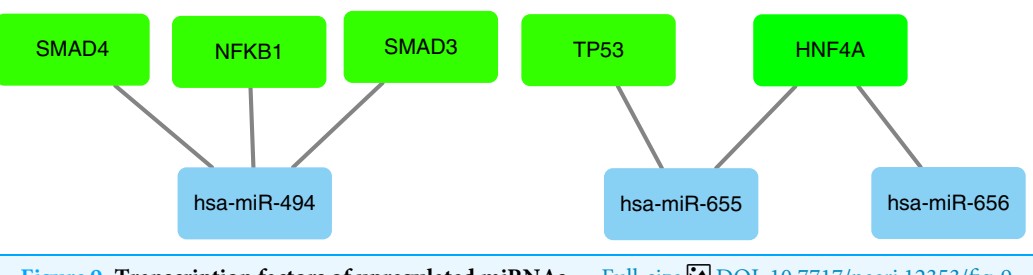

**Figure 9 Transcription factors of upregulated miRNAs.**

target genes (KIF2C, STAT3 and BUB1) were further screened, which was regulated by hsa-miR-494. And in our study, we predicted the upstream transcription factors of hsa-miR-494 were SMAD4, NFKB1, SMAD3. Previous studies have shown that microRNA-494 exerts its effect in cell proliferation, migration, invasion and chemosensitivity in gastric cancer, colorectal cancer, pancreatic cancer, prostate cancer and so on (*Wan, Cheng & Zhang, 2019*; *Ghorbanhosseini et al., 2019*; *Peng et al., 2018*; *He et al., 2018*; *Xu et al., 2018*; *Zhang et al., 2018b*; *Shen et al., 2014*). And one study about ovarian cancer concluded that microRNA-494 is a potential prognostic marker and inhibits cellular proliferation, migration and invasion by targeting SIRT1 (*Yang et al., 2017*).

Among the overlapping three target genes, STAT3 is the most widely studied. STAT3, that is signal transducer and activator of transcription factor 3, relates not only to the proliferation, invasion, metastasis, apoptosis of ovarian cancer cells, but also chemotherapy resistance (*Zhong et al., 2021*; *Yu et al., 2020*; *Liu et al., 2021*; *Pan et al., 2020*).

BUB1 (budding uninhibited by benzimidazole 1) is a mitotic checkpoint protein that is overexpressed in various types of cancer, including breast cancer, gastric cancer, pancreatic cancer and so on (*Han et al., 2015*; *Piao et al., 2019*; *Bai et al., 2019*). Its high expression correlates with a poor clinical prognosis. In a bioinformatics study, BUB1 was reported to be overexpressed in ovarian cancer compared to normal ovarian tissue (*Yang et al., 2020*). Another study confirmed that BUB1 was widely expressed in primary and metastatic ovarian cancer at the level of mRNA and protein (*Davidson et al., 2014*). In our study, we find a new mechanism of BUB1. It may be correlated with cisplatin resistance in ovarian cancer.

KIF2C (Kinesin family member 2C), a member of kinesin superfamily proteins (KIFs), plays important roles in intracellular transport of organelles and maintenance of spindle assembly during mitosis and meiosis. It has been suggested to play an important role in carcinogenesis. Up-regulation of KIF2C has been documented in multiple human cancers, such as breast cancer, gastric cancer, colorectal cancer, glioma, liver cancer and non-small cell lung cancer (*Wei et al., 2020*; *Gan et al., 2019*; *Li et al., 2020*; *Bie et al., 2012*; *Ishikawa et al., 2008*; *Nakamura et al., 2007*). One study reports that KIF2C may be involved in the development of paclitaxel resistance in ovarian cancer (*Zhao et al., 2014*). In our study, the expression of KIF2C was lower in platinum-resistant ovarian cancer cell lines.

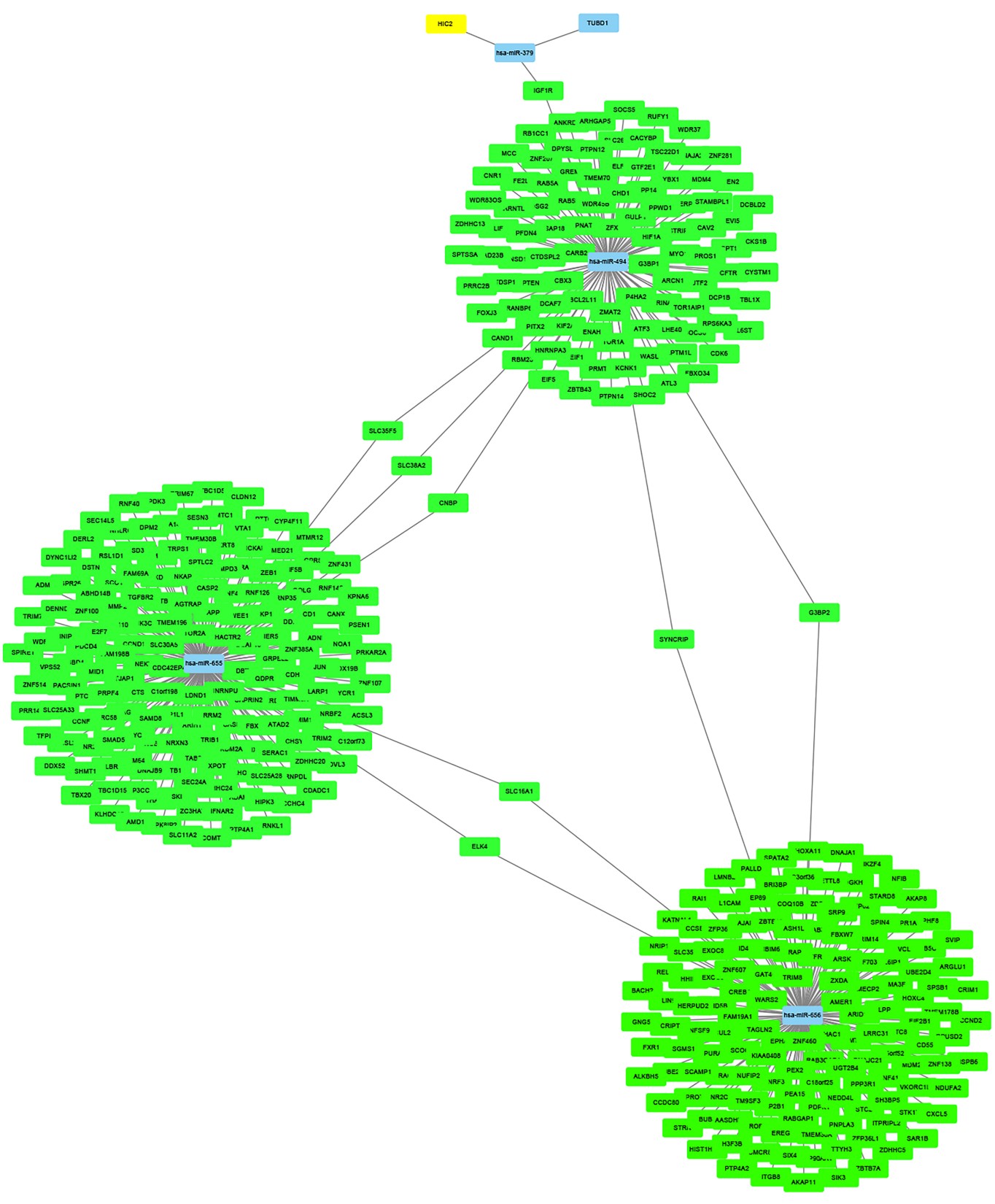

**Figure 10 The miRNA-target gene regulatory network.**

Although we constructed a potential TF-miRNA-mRNA regulatory network in this study, some limitations still exist. First, the dataset GSE28739 in this study included a small sample size of data. Second, lack of experimental validation of our predicted TF-miRNA-mRNA regulatory network is another limitation of the study. Therefore, experimental studies should be executed to validate the TF-miRNA-mRNA regulatory network in the future. And the mechanism of platinum resistance of these genes in ovarian cancer will be reported in the future.

## CONCLUSIONS

In conclusion, we constructed a potential TF-miRNA-mRNA regulatory network related to platinum resistance in ovarian cancer. It can help us identify some key genes and pathways concerned with platinum resistance in ovarian cancer, which may assist in the treatment of ovarian cancer and improve prognosis of ovarian cancer patients by targeting the established miRNA-mRNA regulatory network in the future.

## ACKNOWLEDGEMENTS

We thank Professor Li Bin from National Cancer Institute, Cancer Hospital, Chinese Academy of Medical Sciences, Peking Union Medical College for their gifts of ovarian cancer cell lines. We also thank the GEO, DAVID, KEGG, STRING, miRNet and Cytoscape for providing their platforms and the contributors for their valuable data sets.

### Funding

This research was funded by the Beijing-Tianjin-Hebei Basic Research Cooperation Project (No. J200009 20JCZXJC00010). The funders had no role in study design, data collection and analysis, decision to publish, or preparation of the manuscript.

### Grant Disclosures

The following grant information was disclosed by the authors:
Beijing-Tianjin-Hebei Basic Research Cooperation: J200009 20JCZXJC00010.

### Competing Interests

The authors declare that they have no competing interests.

### Author Contributions

- Wenwen Wang performed the experiments, analyzed the data, prepared figures and/or tables, authored or reviewed drafts of the paper, and approved the final draft.
- Wenwen Zhang performed the experiments, authored or reviewed drafts of the paper, and approved the final draft.
- Yuanjing Hu conceived and designed the experiments, analyzed the data, authored or reviewed drafts of the paper, and approved the final draft.

## Data Availability

The raw data are available as Supplemental Files.

## Supplemental Information

Supplemental information for this article can be found online at http://dx.doi.org/10.7717/peerj.12353#supplemental-information.

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
