# Peer review of "Identification of keygenes, miRNAs and miRNA-mRNA regulatory pathways for chemotherapy resistance in ovarian cancer"

_PeerJ, doi:10.7717/peerj.12353_

## Round 0.1 · original submission · Major Revisions

The manuscript you submitted to PeerJ, has been reviewed. The reviewers have recommended publication pending major revisions. Therefore, I invite you to respond to the reviewers' comments at the bottom of this letter and revise your manuscript accordingly.

Reviewer 1 ·

Basic reporting

Good

Experimental design

Author must include construction and analysis of target genes - TF regulatory network and target genes - miRNA regulatory network along with topology Tables

Validity of the findings

good

Additional comments

Author must provided Differential gene expression (DEGs) table with probe id, logFC, pValue, adj.P.Val, t value and Gene Name, which are more fundamental and basic in this work.
Author must include construction and analysis of target genes - TF regulatory network and target genes - miRNA regulatory network along with topology Tables

Reviewer 2 ·

Basic reporting

none

Experimental design

none

Validity of the findings

none

Additional comments

The presented work in this article is incomplete and not clear.

Reviewer 3 ·

Basic reporting

- English language should be improved significantly. There still have some unclear and ambiguous parts.

- This manuscript lacks a literature review on bioinformatics-based ovarian cancer studies.

- Quality of figures should be improved.

Experimental design

- A critical concern is the use of a very small sample size of data (only 25 patients). This number was not enough to convince a significant result/finding.

- There must have more description of the methodology to support replicating the methods. Currently, it lacks detailed information.

Validity of the findings

- The authors did not have any validation data.

- What are the cut-off threshold and p-values of GO enrichment analysis?

- What are the limitations of the study and how to address them in future works?

Additional comments

No comment.

---

## Round 0.2 · accepted · Accept

Thanks for the revision of the manuscript, which can be now accepted.

Reviewer 3 ·

Basic reporting

No comment.

Experimental design

No comment.

Validity of the findings

No comment.

Additional comments

No comment.